# Early Prediction of Dementia Using Feature Extraction Battery (FEB) and Optimized Support Vector Machine (SVM) for Classification

**DOI:** 10.3390/biomedicines11020439

**Published:** 2023-02-02

**Authors:** Ashir Javeed, Ana Luiza Dallora, Johan Sanmartin Berglund, Alper Idrisoglu, Liaqat Ali, Hafiz Tayyab Rauf, Peter Anderberg

**Affiliations:** 1Aging Research Center, Karolinska Institutet, 171 65 Stockholm, Sweden; 2Department of Health, Blekinge Institute of Technology, 371 79 Karlskrona, Sweden; 3Department of Electrical Engineering, University of Science and Technology Bannu, Bannu 28100, Pakistan; 4Centre for Smart Systems, AI and Cybersecurity, Staffordshire University, Stoke-on-Trent ST4 2DE, UK; 5School of Health Sciences, University of Skövde, 541 28 Skövde, Sweden

**Keywords:** dementia, feature fusion, machine learning, imbalance classes

## Abstract

Dementia is a cognitive disorder that mainly targets older adults. At present, dementia has
no cure or prevention available. Scientists found that dementia symptoms might emerge as early as
ten years before the onset of real disease. As a result, machine learning (ML) scientists developed
various techniques for the early prediction of dementia using dementia symptoms. However, these
methods have fundamental limitations, such as low accuracy and bias in machine learning (ML)
models. To resolve the issue of bias in the proposed ML model, we deployed the adaptive synthetic
sampling (ADASYN) technique, and to improve accuracy, we have proposed novel feature extraction
techniques, namely, feature extraction battery (FEB) and optimized support vector machine (SVM)
using radical basis function (rbf) for the classification of the disease. The hyperparameters of SVM are
calibrated by employing the grid search approach. It is evident from the experimental results that the
newly pr oposed model (FEB-SVM) improves the dementia prediction accuracy of the conventional
SVM by 6%. The proposed model (FEB-SVM) obtained 98.28% accuracy on training data and a testing
accuracy of 93.92%. Along with accuracy, the proposed model obtained a precision of 91.80%, recall of
86.59, F1-score of 89.12%, and Matthew’s correlation coefficient (MCC) of 0.4987. Moreover, the newly
proposed model (FEB-SVM) outperforms the 12 state-of-the-art ML models that the researchers have recently presented for dementia prediction.

## 1. Introduction

Dementia is a mental disorder marked by a progressive decline in cognitive functions that interferes with daily living skills such as visual perception, problem-solving, memory, and the ability to focus on a single topic. Dementia is more common in older adults, yet many consider it an inevitable outcome of aging. This impression of aging might be incorrect. A wide range of illnesses and injuries to the brain are the primary causes of dementia development [1]. The number of people with dementia is swiftly growing globally, and statistical forecasts indicate that 135 million individuals might have dementia by 2050 [2]. According to the World Health Organization, dementia is the sixth leading cause of death globally, and it is the leading cause of disability and dependency among the aged worldwide [3].

Current early-stage dementia diagnosis relies on pathology features or cognitive diagnostic procedures. Pathology features can be detected via neuroimaging. Magnetic resonance imaging (MRI) is employed to investigate the changes in neuron structure [4]. Electroencephalography (EEG) is utilized to assess event-related possibilities to determine the early stages of dementia [5]. However, such techniques are ineffective in identifying dementia. These prediction tests are relatively inexpensive and time-consuming. In addition, a recent study proposes employing computed tomography (CT) or electromagnetic resonance imaging (MRI) to rule out structural causes of the clinical phenotype [6]. It is estimated that primary-care physicians overlook between 29% and 76% of dementia patients or suspected dementia patients [7]. The use of cognitive tests to assess the early stages of dementia also has some disadvantages. It is difficult for paramedics to engage patients and urge them to participate in testing procedures because older individuals often fear visiting clinics. On the other hand, dementia tests performed at home are usually conducted by inexperienced relatives unfamiliar with the scales; as a result, many test findings are incorrect. ML algorithms provide a novel approach to this challenge. Because of developments in information technology, paramedics now have better access to patients’ lives and can detect decreased cognitive function at an earlier stage. ML algorithms may also provide expert medical knowledge. ML-based tools can provide excellent accuracy and a user-friendly approach to the early prediction of dementia. Scientists have built multiple automated diagnostic systems for various ailments using ML methods, such as heart failure [8,9,10], Parkinson’s [11], hepatitis [12], and clinical decision support systems [13].

Different automated diagnostic methods based on ML methodologies have been presented in the past for early diagnosis of dementia. F. A. Salem et al. developed an ML algorithm for dementia prediction using a regression model. They also studied ML techniques for unbalanced classes in the dataset. They utilized oversampling and undersampling to eliminate the bias in the ML model. The balanced random forest (RF) model was the most resilient probabilistic model using only 20 variables from the dataset. Their proposed model reported an F1-score of 0.82%, G-Mean of 0.88%, and AUC of 0.88% [14]. Dallora et al. [15] employed decision trees (DT) to evaluate predictive factors for the ten-year prediction of dementia. In their proposed technique, they employed a recursive feature elimination (RFE) feature selection method to determine the most important variables from a dataset for dementia classification. Their proposed approach, based on RFE and DT, had an AUC of 74.50%. Through feature engineering and genetic algorithms, F.G. Gutierrez et al. devised an ML approach for diagnosing AD and FTD. Their suggested method had an accuracy of 84% [16]. G. Mirzaei and H. Adeli investigated cutting-edge ML approaches for identifying and categorizing AD [17]. H. Hsiu et al. investigated ML techniques for the early detection of cognitive deterioration using a threefold cross-validation approach, and their suggested model achieved an accuracy of 70.32% [18]. A. Shahzad et al. proposed an ML model for pre-screening MCI using an inertial sensor-derived gait biomarker with a 71.67% accuracy rate [19].

### Aim of Study

The previously proposed ML models suffer from lower accuracy and bias in ML models. Motivated by these factors, we proposed a novel feature extraction method to extract useful features from the dataset. Moreover, we optimized an SVM using a grid search algorithm. The proposed hybrid model uses two components: the feature extraction battery (FEB) and support vector machine (SVM), leading to the newly proposed model, namely, (FEB-SVM). To address the issue of bias in the ML model, we deployed the adaptive synthetic sampling (ADASYN) scheme to balance the classes in the dataset. To validate the effectiveness of the proposed model (FEB-SVM), we used different evaluation metrics, e.g., accuracy, precision, recall, PR curve, area under the curve (AUC), F1 score, and Matthew’s correlation coefficient (MCC). Moreover, we conducted three different experiments to evaluate the performance of the newly proposed model.

It is essential to understand that dementia is classified into various subtypes, the most frequent being AD and Vascular dementia, among others. Mixed comorbidities are uncommon, and AD commonly comports with Vascular or Lewy Bodies dementia. Furthermore, unusual forms of AD are sometimes misdiagnosed, according to [20]. The study mentioned here makes no difference between subtypes of dementia, and the word dementia refers to all kinds of dementia.

## 2. Materials and Methods

This research collected data from the Swedish National Study on Aging and Care (SNAC) for the experimental purpose of the proposed model (FEB-SVM). The SNAC is a longitudinal consortium that has been collecting multimodal data from the Swedish senior population to “create trustworthy, comparable, durable datasets” to be used for aging research and aged care [21]. The SNAC was developed as a multifunctional program to explore healthcare quality for the aging population. It comprises a database containing details regarding physical assessment, metacognition, social variables, lifestyle factors, medical records, and so on. Blekinge, Skåne, Nordanstig, and Kungsholmen are the sites from which the SNAC database is collected. They consider a couple of Swedish counties—municipal and borough. This research adopted the SNAC-Blekinge baseline examination, with data gathered from 2000 to 2003. In the literature, there is substantial evidence that environmental factors may impact dementia development [22,23]. This research is based on standard criteria and uses data from urban areas (Blekinge). The inclusion methodology used to eliminate individuals from this investigation is given as follows:Individuals who presented a dementia diagnosis at the beginning of the study or before the ten-year mark.Individuals with missing data in the outcome column.Individuals with more than 10% incomplete data.Individuals who expired (died) well before the ten-year study period.

From the 1402 participants in the SNAC-Blekinge baseline, after the application of the selection criteria, 726 participants (313 males and 413 females) were included, 91 (12.5%) of whom had dementia in the 10 years, and 635 (87.5%) who were free of dementia. The demographics of the sample population in the collected dataset are shown in Table 1. The factors chosen from the SNAC-Blekinge database were based on published research [24,25]. The collected dataset (SNAC-Blekinge) consists of 13 physical measurement parameters such as body mass index (BMI), pain in the last four weeks, heart rate sitting, heart rate lying, blood pressure on the right arm, hand strength in the right arm, hand strength in the left arm, feeling of safety when rising from a chair, assessment of rising from a chair, single-leg standing with right leg, single-leg standing with left leg, dental prosthesis, and several teeth.

It is important to remember that all of the features used in the SNAC were picked based on the evidence relevant to the aging process [21]. At the commencement of the study (2000–2003), 75 variables were identified from the seven categories: demographics, lifestyle, social, psychological, medical history, physical examination, blood tests, and the assessment of various health instruments connected to dementia examination. Medical practitioners provide the target variable employed by the proposed model to predict dementia 10 years after the SNAC baseline. The dementia diagnosis was made using the International Statistical Classification of Diseases and Related Health Problems-10th Revision (ICD-10) and the Diagnostic and Statistical Manual of Mental Disorders (DSM-IV). Table 2 provides a feature category and name from the selected dataset (SNAC-Blekinge).

### Proposed Work

In this work, we designed an automated diagnostic technique for the early prediction of dementia using machine learning and data mining approaches. The suggested diagnostic system is divided into two modules: the first module extracts valuable features from datasets to avoid the problem of model overfitting and the second module works as a classifier to predict dementia. We developed a novel feature extraction method based on linear discriminate analysis (LDA), independent component analysis (ICA), principal component analysis (PCA), locally linear embedding (LLE), and t-distributed Stochastic Neighbor Embedding (t-SNE). The aforementioned feature extraction methods have been cascaded into a single component, which we named a “feature extraction battery” (FEB). Feature extraction begins with an initial set of measured data. It creates derived values (features) that are meant to be useful and non-redundant, allowing future learning and generalization phases and, in some situations, leading to improved human interpretations. Feature extraction helps reduce the dimensionality of the dataset, which eventually reduces the computational complexity of the machine-learning models. The extracted features from FEB are fed into the predictive module of the proposed diagnostic system for the prediction of dementia. We employed a support vector machine (SVM) as a predictive module, and the working of the proposed diagnostic system (Figure 1).

The first stage of the proposed diagnostic system is data preprocessing because data play a vital role in predictive ML models. The dataset is refined, standardized, and normalized. We deal with the missing values in the data preprocessing stage by employing K-nearest neighbors (KNN) imputation [26]. This technique finds the K-items comparable (near) to the missing data. The KNN replaces the mean or most common value of K in the missing data with the most comparable neighbors. The selected dataset for the experiments portrays highly imbalanced classes. Hence, KNN imputation is employed independently on missing data from the majority and minority classes. Through this technique, the chance of infecting the minority class with data from the dominant class was reduced. Following the resolution of missing values, we performed the StandardScaler function on the selected dataset. The StandardScaler function helps to standardize a dataset by eliminating the mean and scaling to unit variance. A sample’s average score λ is computed as follows:(1)$$Z=\frac{\left(\lambda -\upsilon \right)}{\gamma }$$
where υ denotes the mean of training samples and γ is the standard deviation of the training samples. By calculating the relevant statistics on the training set samples, standardizing and scaling are performed independently on each feature. The mean and standard deviation are then saved for further use to transform on additional data.

When ML models are trained using all the feature space of a dataset, they tend to overfit, which means that ML models display improved performance on training data but poor performance on testing data [27,28]. This might be because the classifier learned superfluous or noisy features in the training data, or it could be due to a weak classifier with too many parameters. As a result, we should extract a subset of features from the dataset and a properly constructed classifier. In feature extraction methods, new features are constructed from the given dataset. Feature extraction decreases the resources necessary to explain a vast dataset. One of the primary issues with analyzing complex data is the number of variables involved. Analysis with many variables often necessitates a substantial memory and computing capacity. It may also lead a classification method to overfit training examples and generalize poorly to new samples. Feature extraction is a broad term encompassing ways of building variable combinations to avoid these issues while accurately summarizing the data. Many machine learning practitioners feel that well-optimized feature extraction is the key to good model design [29]. Therefore, we proposed a novel feature extraction method (FEB) to avoid the problem of model overfitting while simultaneously reducing data dimensionality. Reduced data dimensionality increases the performance of the proposed SVM-FEB in terms of time complexity. In FEB, we cascaded different feature extraction methods (LDA, PCA, ICA, LLE, TSNE) into a single module. The four feature extraction methods (PCA, ICA, LLE, TSN) construct a couple of new features, while LDA constructs only one new feature. The newly extracted FEB features are combined to generate an optimum dataset with low dimensionality. The proposed FEB constructs nine new features from the original dataset, which consists of 75 features.

After the feature extraction stage, we divided the dataset into two parts; one for training and the other for the testing purposes of the proposed SVM-FEB model. The classes in the dataset for the experiments are highly imbalanced, which means that the model would be biased toward the majority class. To address this issue, we used the ADASYN approach to tackle the imbalanced class issue [30]. The ADASYN approach uses a density distribution $\delta_{i}$ as a criterion to automatically compute the number of synthetic samples necessary for minority data samples. $\delta_{i}$ is a physical evaluation of the weight distribution of unique minority class instances depending on their level of learning difficulty. Following ADASYN, the final dataset will not only provide a balanced structure of classes in the data distribution (as defined by the coefficient) but also compels the learning algorithm to concentrate on complicated cases. As a consequence, the proposed system (FEB-SVM) is trained on balanced data, mitigating the risk of bias in the ML model. It is worth noting that the ADASYN approach is applied to training data following the data split. Suppose the ADASYN technique is used for the entire dataset (i.e., before data partitioning). In that case, the performance of the ML model will be skewed because samples from the testing dataset will also be included in the training dataset. Using ADASYN to balance the training dataset, we employed SVM for the classification task.

SVM is a powerful tool for classification and regression problems [31]. SVM attempts to construct a hyperplane with the greatest possible margin. In the case of a classification problem, the hyperplane $h\left(x\right)=\left(\sigma^{T}*x\right)+\gamma$, where γ denotes the bias and σ represents a weight vector that is built using training data and serves as a decision boundary for determining the class of a data point (a multidimensional feature vector). In the case of binary classification, SVM employs two support vectors and identifies the nearest vectors (data points) of two classes to create a margin. These vectors are referred to as support vectors. Margin is computed by taking the perpendicular distance between the lines going through the support vectors and multiplying by $\frac{2}{\left||\sigma \right|{|}_{2}^{2}}$. The primary objective is to develop an optimized SVM predictive model to provide an ideal hyperplane with the highest margin. SVM employs a set of slack variables known as $\nu_{i}$, i = 1, 2, …, ℧, as well as a penalty parameter known as β, and attempts to maximize ${\left||\sigma |\right|}_{2}^{2}$ while minimizing misclassification errors. This fact is mathematically expressed as follows:(2)$$\underset{Regularized}{\underset{︸}{\min_{\sigma ,\gamma ,\nu }\frac{1}{2}{∥\sigma ∥}_{2}^{2}}}+\gamma \underset{Error}{\underset{︸}{\sum_{i=1}^{℧}\nu_{i}}}$$
(3)$$S.t\left\{\begin{array}{cc} y_{i}\left(\sigma x_{i}+\gamma \right)\geq 1-\nu_{i} & \\ \nu_{i}\geq 0,i=1,2,\ldots ,℧ \end{array}\right.$$
where ν is the slack variable that is used to calibrate the degree of misdiagnosis and penalized factor is the Euclidean norm, also known as the L2-norm.

The major difficulty is that a linear hyperplane cannot correctly partition the binary classes’ data points (i.e., with the lowest classification error). For this reason, the SVM employs a kernel technique in which the SVM model converts local data points into hyperdimensional points to convert non-separable data points into separable data points. Different kernels are used, including the radial basis function (RBF) kernel, linear kernel, sigmoid kernel, and polynomial kernel. These kernels are SVM model hyperparameters that must be adjusted for each task. To design the SVM model that works best on a dementia prediction challenge, we must carefully update or optimize its hyperparameters. Grid search is the main way to reach this purpose. As a result, we employed the grid search approach to tweak the SVM hyperparameters. Consequently, in this paper, we suggest an FEB decrease in the data dimensionality. The suggested SVM-FEB approach dynamically optimizes the SVM model’s hyperparameters using the grid search method.

## 3. Validation and Evaluation

The holdout validation technique has been extensively used in the literature as a benchmark for assessing the effectiveness of ML-based diagnostic systems [10,12,13]. In a holdout validation procedure, a dataset is partitioned into a couple of segments where one half is utilized for training while the remaining half is utilized for testing purposes of the proposed ML model. The dataset is split with a ratio of 70% for training the ML model and 30% used for testing. Thus, in our tests, we employed the data mentioned above partition criteria for the training and testing of the proposed SVM-FEB model. Following the data partition, we select evaluation measures to compare the performance of the proposed model with existing state-of-the-art ML models for dementia prediction. The assessment criteria used for evaluating the SVM-FEB model are accuracy, precision, recall, F1-score, Matthew’s correlation coefficient (MCC), and area under the curve (AUC) using PR plot. The evaluation metrics are presented mathematically as
(4)$$Accuracy=\frac{T^{+}+T^{-}}{T^{+}+T^{-}+F^{+}+F^{-}}$$
where $T^{+}$ stands for the No. of true positive, $F^{+}$ represents the No. of false positives, $T^{-}$ denotes the number of true negatives, and $F^{-}$ represents the No. of false negatives.
(5)$$Precision=\frac{T^{+}}{T^{+}+F^{+}}$$
(6)$$Recall=\frac{T^{+}}{T^{+}+F^{-}}$$
(7)$$F1\_score=\frac{2T^{+}}{2T^{+}+F^{+}+F^{-}}$$
(8)$$MCC=\frac{T^{+}\times T^{-}-F^{+}\times F^{-}}{\sqrt{\left(T^{+}+F^{+}\right)\left(T^{+}+F^{-}\right)\left(T^{-}+F^{+}\right)\left(T^{-}+F^{-}\right)}}$$

A binary classification problem is statistically examined. The F1-score is defined as the F-measure. The F1-score gives a score between 0 and 1, with 1 representing excellent predictions and 0 representing the worst. MCC is used to determine whether or not a test is correct. The value range for MCC is between 1 and −1, with 1 being the best prognosis and −1 representing the worst prediction.

## 4. Experimental Results

Three kinds of experiments were conducted on the dementia dataset to examine the efficacy of the newly developed model (FEB-SVM). The first experiment used the grid search algorithm to construct and optimize traditional ML models using all the dataset’s features (75). The proposed SVM-FEB approach is built in the second experiment. At the same time, additional state-of-the-art ML models are constructed in the third experiment while utilizing the same dementia data and a novel, suggested feature extraction module (FEB). All computation tasks were carried out on an Intel (R) Core (TM) i5-8250U CPU running at 1.60 GHz using Windows 10 64bit. The Python software package is employed to carry out all of the experiments.

### 4.1. Experiment No.1: Performance of ML Models Using All Features

In this experiment, we set up multiple ML models, i.e., naive Bayes (NB), logistic regression (LR), decision tree (DT), random forest (RF), k-nearest neighbors (KNN), and support vector machine (SVM) with various kernels (rbf, linear, polynomial, sigmoid), which are implemented in Python. The efficiency of the constructed ML models was assessed using all the dataset’s features (75). It is noteworthy that the classes in the dataset are balanced through the ADASYN technique. Table 3 shows the accuracy, precision, recall, and MCC of dementia prediction. The SVM with a polynomial kernel obtained the best dementia diagnosis accuracy of 88.59 percent. However, the SVM training accuracy is lower than the test accuracy, indicating that the ML model overfits. Therefore, we constructed a feature extraction battery (FEB) to prevent the problem of model overfitting.

### 4.2. Experiment No.2: Performance of Proposed Model SVM-FEB

In the proposed model (FEB-SVM), features are extracted from the dataset using a novel feature extraction battery (FEB). From FEB, we obtained the nine features that were extracted from the whole dataset. The classes were found to be highly unbalanced; so, we deployed the adaptive synthetic (ADASYN) oversampling method. After balancing the classes in the dataset, we optimized the hyperparameters of the SVM with the rbf kernel. We employed a grid research algorithm for the optimization of SVM hyperparameters. The optimal values of the hyperparameters of SVM are set by exploiting an exhaustive grid search algorithm. Table 4 shows that the proposed model SVM-FEB achieved the best dementia prediction accuracy, 93.92%, where C = 10 and G = 0.1 (C = Cost, G = Gamma) are the values of hyperparameters that are searched through using the grid research technique. The comparison of Table 3 and Table 4 shows that the proposed model SVM-FEB improves the performance of traditional SVM by 8%. Furthermore, traditional SVM with rbf used all the features of the dataset (see Table 3), while in the proposed model SVM-FEB, only 9 features are used for the prediction of dementia (see Table 4).

To validate the efficiency of the proposed model (FEB-SVM), we also used a precision–recall (PR) curve. In total, two types of experiments were conducted. In the first experiment, a simple SVM with an rbf kernel is tested against the PR-curve; in contrast, in the second experiment, the proposed SVM-FEB is tested. The important parameter in the PR plot is the area under the curve (AUC), in which a model having more area under the curve is considered better. Figure 2b shows the PR curve plot of the traditional SVM model with an AUC of 88% while Figure 2a presents the PR plot of the proposed SVM-FEB model with an AUC of 93%. Hence, the proposed model has better dementia prediction accuracy in comparison to the simple SVM model.

Furthermore, we analyzed the training and testing accuracy of the proposed model (SVM-FEB) with other ML models. From Figure 3, it can be observed that the proposed (SVM-FEB) model achieved the highest testing accuracy compared with the rest of the ML models.

### 4.3. Experiment No.3: Performance of ML Models Based on FEB

In this experiment, we compared the newly developed model (SVM-FEB) performance with other ML models using a novel feature extraction model (FEB). We selected the following ML models, i.e., Naive Bayes (NB), K-Nearest Neighbors (KNN), Random Forest (RF), Decision Tree (DT), Logistic Regression (LR), and Support Vector Machine (SVM) with different kernels (linear, sigmoid, ref, linear). The selected ML models’ hyperparameters were optimized using a grid search algorithm. For a fair comparison, we used balanced classes in the dataset, which were obtained through the ADASYN technique. Table 5 presents the results of each ML model along with the values of tuned hyperparameters, i.e., D—depth, E—edge, Ne—number of estimators, G—gamma, and K—number of neighbors. The performance of each model is assessed across different evaluation metrics such as accuracy on training data (ACC._train), accuracy on test data (ACC._test), precision, recall, f1_score, and Matthew’s correlation coefficient (MCC). The proposed model (SVM-FEB) has achieved the highest accuracy of 93.92% in comparison with other ML models using the same feature extraction module (FEB), as shown in Table 5. Furthermore, we can also compare the results of Table 5 with Table 3 where the same ML models are used using all dataset features. From the comparative analysis of both tables, it can be seen that the novel proposed feature extraction method (FEB) has significantly improved the performance of ML models. Thus, the proposed feature extraction model reduced the complexity of the ML models because ML models used only 9 features compared with all the available feature space of 75 features. The proposed feature extraction module (FEB) and optimized SVM with an rbf kernel obtained improved accuracy results compared with conventional SVM.

### 4.4. Comparison of Dementia Prediction Methods

We evaluated the performance of numerous ML-based automated diagnostic systems that were proposed by researchers in the literature for dementia prediction. Table 6 summarizes the performance of previously proposed ML-based approaches for dementia prediction along with our proposed model. Compared with the recently proposed models such as F. A. Salem et al. [14], F. G. Gutierrez et al. [16], G. Mirzaei and H. Adeli [18], and A. Shahzad et al. [19] and A. Javeed et al. (2022) [32], our newly developed model (FEB-SVM) performed considerably better.

## 5. Conclusions

In this study, we addressed the problems of low accuracy and bias in ML models for dementia prediction, which researchers in the recent past have raised. Unfortunately, dementia is a rare, occurring disease, so classes in the dementia datasets are significantly imbalanced, causing bias in ML models. Therefore, we deployed the adaptive synthetic sampling (ADASYN) technique to resolve this issue. For improved accuracy, we proposed a novel feature extraction (FEB) method that extracts the valuable features from the dataset so that the proposed model would not learn the noisy features from the dataset and avoid model overfitting problems. The feature extraction (FEB) module extracts 9 features from the dataset, comprising 75 features. The FEB helps to improve the accuracy and reduce the computational complexity of the proposed model. For classification purposes, SVM is deployed with different kernel functions, and hyperparameters of the SVM are fine-tuned using a grid research algorithm. From the experimental results, it is evident that the proposed model (FEB-SVM) improved the performance of conventional SVM by 6% for dementia prediction. Moreover, the proposed model outperformed the 12 recently proposed models based on ML for dementia prediction, which the researchers presented. However, the proposed method (SVM-FEB) has significant shortcomings that the researchers must overcome. The newly proposed FEB’s constructed features cannot help identify the features causing dementia in older persons because the newly generated FEB is based on feature extraction techniques. Therefore, new methods should be proposed in the future for identifying the features that cause dementia problems based on feature selection algorithms [39]. Furthermore, a machine learning model based on meta-heuristics and deep learning should be built to improve dementia prediction accuracy.

## Figures and Tables

**Figure 1 biomedicines-11-00439-f001:**
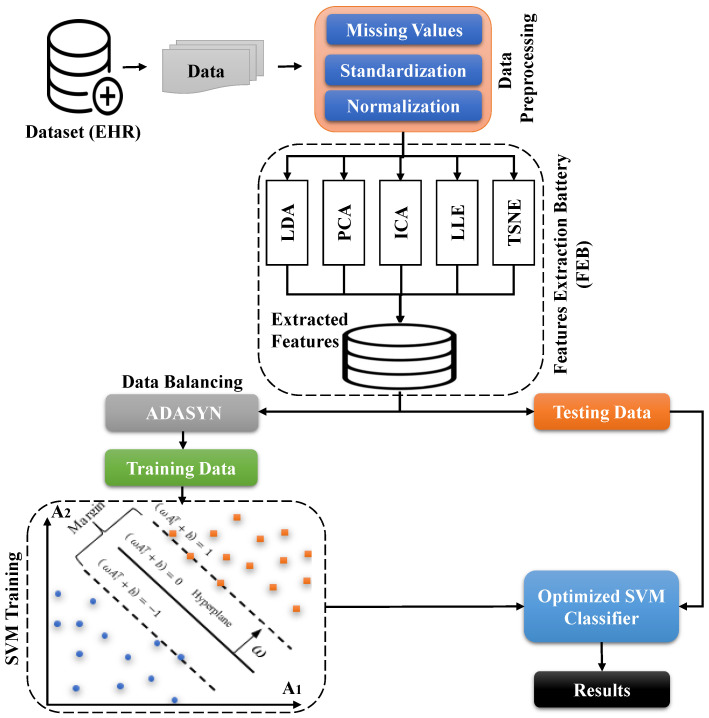
Schematic overview of the proposed model SVM-FEB.

**Figure 2 biomedicines-11-00439-f002:**
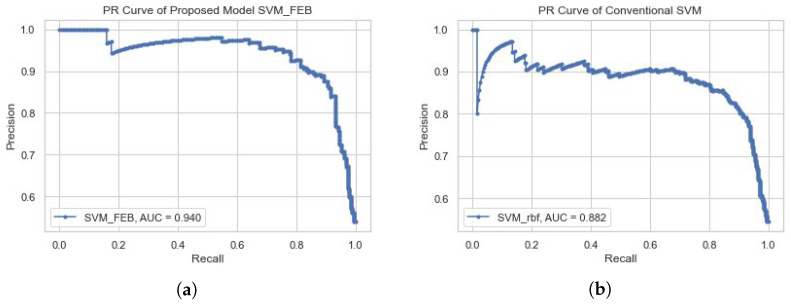
Performance comparison based on AUC. (**a**) PR curve of proposed model; (**b**) PR curve of the conventional SVM.

**Figure 3 biomedicines-11-00439-f003:**
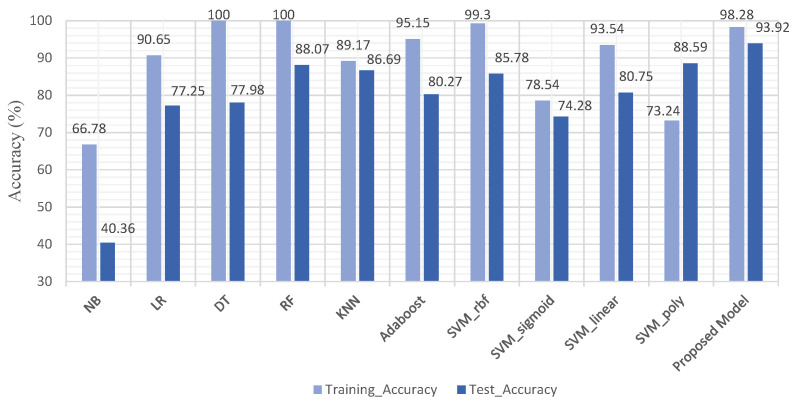
Performance comparison of proposed model with other ML models.

**Table 1 biomedicines-11-00439-t001:** Demographic overview of the data samples.

Age_Group	Male	Female	Healthy_Subject	Dementia_Cases
60	82	82	164	02
66	75	95	170	06
72	50	74	124	10
78	41	50	91	17
81	35	46	81	19
84	26	42	68	22
87	04	19	23	14
90+	00	05	05	01
Total	313	413	726	91

**Table 2 biomedicines-11-00439-t002:** Overview of features in the dataset.

Feature_Class	Feature_Names	Total
Demographic	Age, Gender	02
Lifestyle	Light Exercise, Alcohol Consumption, Alcohol Quantity, Work Status, Physical Workload, Present Smoker, Past Smoker, Number of Cigarettes a Day, Social Activities, Physically Demanding Activities, Leisure Activities	11
Social	Education, Religious Belief, Religious Activities, Voluntary Association, Social Network, Support Network, Loneliness	07
Physical Examination	Body Mass Index (BMI), Pain in the last 4 weeks, Heart Rate Sitting, Heart Rate Lying, Blood Pressure on the Right Arm, Hand Strength in Right Arm in a 10 s Interval, Hand Strength in Left Arm in a 10 s Interval, Feeling of Safety from Rising from a Chair, Assessment of Rising from a Chair, Single-Leg Standing with Right Leg, Single-Leg Standing with Left Leg, Dental Prosthesis, Number of Teeth	13
Psychological	Memory Loss, Memory Decline, Memory Decline 2, Abstract Thinking, Personality Change, Sense of Identity	06
Health Instruments	Sense of Coherence, Digit Span Test, Backwards Digit Span Test, Livingston Index, EQ5D Test, Activities of Daily Living, Instrumental Activities of Daily Living, Mini-Mental State Examination, Clock Drawing Test, Mental Composite Score of the SF-12 Health Survey, Physical Composite Score of the SF-12 Health Survey, Comprehensive Psychopathological Rating Scale	12
Medical History	Number of Medications, Family History of Importance, Myocardial Infarction, Arrhythmia, Heart Failure, Stroke, TIA/RIND, Diabetes Type 1, Diabetes Type 2, Thyroid Disease, Cancer, Epilepsy, Atrial Fibrillation, Cardiovascular Ischemia, Parkinson’s Disease, Depression, Other Psychiatric Diseases, Snoring, Sleep Apnea, Hip Fracture, Head Trauma, Developmental Disabilities, High Blood Pressure	22
Biochemical Test	Hemoglobin Analysis, C-Reactive Protein Analysis	02

**Table 3 biomedicines-11-00439-t003:** Performance of ML models on balance data using all features.

Model	Acc._train	Acc._test	Precision	Recall	F1_score	MCC	95% CI ^1^
NB	66.78	40.36	56.00	63.00	40.00	0.1695	0.77, 0.87
LR	90.65	77.25	55.00	57.00	77.00	0.1124	0.82, 0.91
DT	100	77.98	49.00	49.00	78.00	0.1157	0.78, 0.88
RF	100	88.07	55.00	51.00	88.00	0.4058	0.85, 0.93
KNN	89.17	86.69	51.00	50.00	87.00	0.3851	0.82, 0.92
Adaboost	95.15	80.27	66.00	51.00	80.00	0.1834	0.83, 0.91
SVM_rbf	99.30	85.78	88.00	56.00	86.00	0.2131	0.84, 0.93
SVM_sigmoid	78.54	74.28	56.00	63.00	74.00	0.1986	0.83, 0.92
SVM_linear	93.54	80.75	58.00	61.00	81.00	0.2042	0.82, 0.91
SVM_poly	73.24	88.59	45.00	50.00	89.00	0.2331	0.83, 0.92

^1^ CI = Confidence Intervals.

**Table 4 biomedicines-11-00439-t004:** Performance of ML models on balance data while utilizing FEB.

Model	Hyper.	Acc._train	Acc._test	Precision	Recall	F1_score	MCC
SVM_rbf	C:100, G:0.1	99.77	90.88	92.39	87.62	89.50	0.4178
SVM_rbf	C:10, G:0.1	97.48	91.16	92.81	86.59	90.00	0.4216
SVM_rbf	C:100, G:0.1	99.88	91.79	92.22	85.56	91.00	0.4487
SVM_rbf	C:10, G:1	98.28	91.83	91.80	86.59	89.12	0.4387
SVM_rbf	C:10, G:1	98.50	92.46	89.67	85.05	90.00	0.4725
SVM_rbf	C:300, G:0.01	95.46	92.02	93.88	87.11	92.00	0.4747
SVM_rbf	C:10, G:1	100	92.70	89.37	95.36	92.00	0.4852
SVM_rbf	C:10, G:0.01	98.74	92.95	93.82	86.08	92.50	0.4810
SVM_rbf	C:100, G:0.1	98.41	93.31	92.34	87.11	93.00	0.4853
SVM_rbf	C:10, G:0.1	98.28	93.92	91.80	86.59	89.12	0.4987
SVM_linear	C: 0.1, G: 01	84.54	82.99	96.15	77.31	85.71	0.3630
SVM_sigmoid	C: 10, G: 0.001	83.23	82.97	96.52	71.64	82.00	0.3359
SVM_poly	C:01, G:01	88.23	84.57	95.75	90.28	84.00	0.3732

**Table 5 biomedicines-11-00439-t005:** Performance of ML models on balance data using FEB.

Model	Hyper.	Acc._train	Acc._test	Precision	Recall	F1_score	MCC
NB	Var:0.006	79.40	75.22	62.00	77.00	75.00	0.3642
LR	C:100, S: newton, p:l2	84.46	78.36	64.00	79.00	78.00	0.3954
DT	D: 03, E:04	87.80	86.00	96.02	74.74	85.00	0.3374
RF	D:10, Ne:100. E:01	97.95	90.15	94.79	84.53	89.00	0.4286
KNN	Lf:01, K:01, P:02	100	90.54	92.98	81.95	87.00	0.4324
Adaboost	Lr:01, Ne: 10	86.77	85.63	79.38	86.76	86.00	0.3567
SVM_Linear	C: 0.1, G: 01	84.54	84.99	96.15	77.31	85.71	0.3630
SVM_Sigmoid	C: 10, G: 0.001	83.23	82.97	96.52	71.64	82.00	0.3359
SVM_Poly	C:01, G:01	88.23	84.57	95.75	90.28	84.00	0.3732
SVM_rbf	C:10, G:0.1	98.28	93.92	91.80	86.59	89.12	0.4987

**Table 6 biomedicines-11-00439-t006:** Comparison of classification accuracies with previously proposed methods for dementia prediction.

Study (Year)	Method	Accuracy (%)	Balancing
P. C. Cho and W. H. Chen (2012) [33]	PNNs	83.00	No
P. Gurevich et al. (2017) [34]	SVM	89.00	Yes
D. Stamate et al. (2018) [35]	Gradient Boosting	88.00	Yes
Visser et al. (2019) [36]	XGBoost+ RF	88.00	No
Dallora et al. (2020) [15]	DT	74.50	Yes
M. Karaglani et al. (2020) [37]	RF	84.60	No
E. Ryzhikova et al. (2021) [38]	ANN + SVM	84.00	No
F. A. Salem et al. (2021) [14]	RF	88.00	Yes
F. G. Gutierrez et al. (2022) [16]	GA	84.00	No
G. Mirzaei and H. Adeli (2022) [18]	MLP	70.32	No
A. Shahzad et al. (2022) [19]	SVM	71.67	No
A. Javeed et al. (2022) [32]	Autoencoder + Adaboost	90.23	Yes
Proposed Model (2023)	FEB + SVM	93.92	Yes

## Data Availability

Data can be available in Appendix A.

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
