# Peer review of "Early Prediction of Dementia Using Feature Extraction Battery (FEB) and Optimized Support Vector Machine (SVM) for Classification"

_biomedicines, 2023, doi:10.3390/biomedicines11020439_

Round 1
Reviewer 1 Report
The key feature of this paper is a feature extraction methods. Authors should explain about it more throughly.
1) Figure 1
You need to redraw this schematic drawing by using bigger fonts. Since this is the core picture, every aspects of the drawing should be recognizable.
2) #164~#168
You need to state more detailed descriptions of FEB. Are 4 extraction methods produce 2 features each? And LDA produces only one feature? What do you mean by 9 new features? Why extract 9 features rather than 13? Internal structures of FEB are essential parts of this paper, so state them thoroughly and logically.
3) #246
NB, LR, DT, RF
You must state the full names for these algorithms. Any abbreviation should be fully explained when it is used at the first time in a paper.
4) #264
C = Cost, G = Gamma
The full name of these coefficients must be stated. (even though they are well-known)
Author Response
Reviewer comments are address in the attached file.

Reviewer 2 Report
This submission paper describes the machine learning methods based on the feature extraction battery to classify the dementia at an early stage. The adaptive synthetic sampling and support vector machine methods were used to improve the classification performance. The following comments should be considered in the revised manuscript:
1) In Section 2, in addition to the demographic context, the description of the SNAC database should be extended to present the details of physiological measurement parameters by medical instruments. It is suggested a list of table in the text.
2) In Section 4, the experimental results should contain the feature analysis results, such as LDA, PCA, ICA, LLE, tSNE. It is necessary to show how many features have been mapped and reduced for further classification.
3) In Page 6, the details of the SVM model parameters should be provided such as number of hidden layer and neurons, which kernels for optimization, how many support vectors after optimization.
4) It is suggested including the 95% confidence interval of each performance parameter in Table 2.
5) In Section 4.4, it is necessary to discuss why the proposed work could outperform the previous studies.
6) The limitations of the study should be fully discussed.
Author Response
Reviewer comments are addressed in the attached file.

Round 2
Reviewer 2 Report
The authors have replied to the review comments, and the revised manuscript is now satisfied for consideration of publication.